# Puzzles and pitfalls involving Haar-typicality in holography

**Ning Bao[1,2]⋆ and Aidan Chatwin-Davies[1]†**

**1** Walter Burke Institute for Theoretical Physics,
California Institute of Technology, Pasadena, United States
**2** Berkeley Center for Theoretical Physics,
University of California, Berkeley, United States

⋆ ningbao75@gmail.com    † aechatwi@gmail.com

## Abstract

**Holographic states that have a well-defined geometric dual in AdS/CFT are not faithfully represented by Haar-typical states in finite-dimensional models. As such, trying to apply principles and lessons from Haar-random ensembles of states to holographic states can lead to apparent puzzles and contradictions. We point out a handful of these pitfalls.**


## 1  Introduction

Typical states and the technique of averaging over ensembles of states are powerful tools in quantum information theory. In high-energy physics, these tools have been a useful component

of quantum gravity studies over the last several years. In particular, the notion of Haar-typical states and Page's theorem are two workhorses of quantum information in quantum gravity.

Roughly speaking, a Haar-typical state is a fully-random quantum state. Given a finite-dimensional Hilbert space $\mathcal{H}$ and any reference state $|\psi_0\rangle \in \mathcal{H}$, a Haar-typical state may be thought of as a realization of the random variable $|\psi(U)\rangle = U|\psi_0\rangle$, where $U$ is a random unitary matrix drawn with uniform probability from the set of all unitary matrices. The uniform probability distribution over the unitaries is the normalized Haar measure over the set of unitary matrices. For convenience, we will just refer to this as "the Haar measure" for the rest of this note.

The Haar measure is a key ingredient in both the statement and proof of Page's theorem [1]. Suppose now that the Hilbert space splits into two subfactors, $\mathcal{H} = \mathcal{H}_A \otimes \mathcal{H}_B$. If $|\psi\rangle$ is a Haar-typical state on $\mathcal{H}$, then Page's theorem essentially says that, with high probability, the reduced state of $|\psi\rangle$ in the $\mathcal{H}_A$ subfactor is very nearly maximally entangled with the part of the state in the $\mathcal{H}_B$ subfactor if the dimension of $\mathcal{H}_A$ is much smaller than that of $\mathcal{H}_B$. Taken together, Page's theorem and Haar typicality are a precise statement of the notion that a small subsystem generically tends to be maximally entangled with its larger complement in a fixed Hilbert space.

Page's theorem has been fruitfully applied to the physics of semiclassical black holes [2–4]. Here, one usually considers a collection of matter that is initially in a pure state and that collapses into a black hole, which over time dissipates via the process of Hawking evaporation [5]. From a quantum-mechanical standpoint, the whole process is modelled as taking place in a single Hilbert space, $\mathcal{H}$, in which a finite number of degrees of freedom are divided up between the black hole and radiation. Of course, the factorization of $\mathcal{H}$ varies from Cauchy slice to Cauchy slice as the black hole evaporates, but if black holes do not destroy information, then the evolution of the total state is unitary and the size of the total Hilbert space must be constant in this model.

While the quantum-gravitational dynamics of black hole degrees of freedom are unknown, for information-theoretic purposes at least, it seems that it is reasonable to model the dynamics in the black hole's Hilbert space by Haar-random unitary evolution (or a 2-design approximation thereof [4]), at least on timescales that are shorter than the rate of Hawking emissions so that the size of the black hole Hilbert space is constant. Therefore, shortly after the black hole forms,[1] the total state in $\mathcal{H}$ is Haar-typical in this model, and so Page's theorem may be used to study the entanglement properties of the state across the changing factorization of $\mathcal{H}$ during the subsequent evolution. For example, this sort of analysis revealed that if you toss a small quantum system into a black hole, then information about its state is rapidly returned via Hawking radiation once the black hole has given up more than half of its degrees of freedom through evaporation [4]. Similar considerations are at the heart of the ongoing debates that surround complementarity [6], the black hole information problem [7], and firewalls [8,9].

However, an area in which Page's theorem and Haar-typicality have significantly less applicability is AdS/CFT [10,11]. For starters, the Haar measure is not even well-defined for a conformal field theory (CFT), whose Hilbert space is infinite-dimensional. One must therefore work with a finite-dimensional approximation of the CFT in order to define the Haar measure. But, even assuming that such a regularization can be made in a satisfactory way, the Haar-typical states of this approximation will be of limited use in modelling holographic states with well-defined geometric gravitational duals because they are too entangled on short scales. If $A$ is a small spacelike region in the boundary, the reduced state on $A$ of a Haar-typical state of the approximate theory will have extensive entanglement entropy that scales like the volume of $A$, per Page's theorem. For a holographic state, when $A$ is small enough, the minimal surface anchored to $\partial A$ only probes the region of the bulk gravitational dual near its asymptotically Anti

---

[1]Or more precisely, one scrambling time after the black hole forms [4].

de Sitter (AdS) boundary. Therefore, the area of the minimal surface, and hence the entropy of the reduced holographic state on *A*, must be sub-extensive in the volume of *A*. Haar-typical states in the CFT approximation are therefore not good models for holographic states.[2]

That measures of holographic states have distinctive structure is well-known in the community. Nevertheless, it is quite easy to momentarily overlook this fact, which can lead to apparent puzzles in holography. Our main goal in writing this short note is to highlight this observation and to point out a handful of potential pitfalls that arise from overextending Page's theorem when it enters into discussions about holography in the literature. In particular, our goal is not to assess the validity of various cutoffs or finite-dimensional models of CFTs, which is an important topic in and of itself. Rather, such techniques being commonplace in both the literature and the verbal lore (including sometimes the haphazard use of Page's theorem without reference to any sort of finite-dimensional regularization), we only aim to identify limitations on typicality-based arguments which must accompany these techniques. We hope that both veterans and novices will find this note to be digestible and pedagogical.

In Section 2, we review the precise definition of Haar-typicality and the precise statement of Page's theorem, and we reiterate the argument for why Haar-typicality is of limited utility for holography in commensurate language. Then, in Section 3, we point out several puzzles and potential pitfalls in the literature. We offer some concluding remarks in Section 4.

## 2 Properties of random states

Let $\mathcal{H}$ be a finite-dimensional Hilbert space with dimension *n* and consider the group of unitary transformations, U(*n*), acting on $\mathcal{H}$.

**Definition 1** *The normalized* Haar measure *on* U(*n*) *is the unique measure $\mu$ measuring subsets of* U(*n*) *such that:*

- $\mu(U\mathcal{S}) = \mu(\mathcal{S}U) = \mu(\mathcal{S})$ *for all* $\mathcal{S} \subset$ U(*n*) *and* $U \in$ U(*n*), *where* $U\mathcal{S} = \{UV \mid V \in \mathcal{S}\}$ *and* $\mathcal{S}U$ *is similarly defined.*

- $\mu(\mathrm{U}(n)) = 1$

From the properties above, it also follows that $\mu$ is non-negative, so the normalized Haar measure on U(*n*) is a uniform probability distribution on the group of unitary transformations as such.

To state Page's theorem, we also need the trace norm, which is defined as $\|T\|_1 = \mathrm{tr}\,\sqrt{T^\dagger T}$ for any linear operator *T* on $\mathcal{H}$. The trace norm gives a good notion of distinctness of states because if $\|\rho - \sigma\|_1 < \epsilon$ for any two density operators (i.e., states) $\rho$ and $\sigma$ on $\mathcal{H}$, then $\|P(\rho - \sigma)\|_1 < \epsilon$ for any projector *P* so that the probabilities for measurement outcomes in the states $\rho$ and $\sigma$ are close [12].

Now suppose that $\mathcal{H} = \mathcal{H}_A \otimes \mathcal{H}_B$, where dim $\mathcal{H}_A = d_A$ and dim $\mathcal{H}_B = d_B$ with $n = d_A d_B$. Without loss of generality, suppose that $d_A \leq d_B$. A precise statement of Page's theorem is as follows:[3]

---

[2]Nevertheless, careful use of random state statistics and suitable generalizations of Haar-typicality, such as typicality with respect to the microcanonical ensemble in a fixed energy window, can serve as useful tools for holography. We will return to this point in section 3.5.

[3]As a historical note, the theorem originally proved by Page [1] was formulated in terms of entanglement entropies and built upon earlier works by Lubkin [13] and Lloyd & Pagels [14]. The version of Page's theorem given here is based on the statement appearing in [12]. A detailed yet accessible proof of Page's theorem can be found in Sec. 10.9 of [15] under the name of "the decoupling inequality."

**Theorem 2 (Page)** *Let $|\psi_0\rangle \in \mathscr{H}$ be a fixed reference state and let $|\psi(U)\rangle = U|\psi_0\rangle$ for any $U \in \mathrm{U}(n)$. Let $\rho_A(U) = \mathrm{tr}_B\,|\psi(U)\rangle\langle\psi(U)|$ denote the reduced state of $|\psi(U)\rangle$ on $\mathscr{H}_A$. Then it follows that*

$$\int d\mu(U) \left\| \rho_A(U) - \frac{I_A}{d_A} \right\|_1 \leq \sqrt{\frac{d_A}{d_B}} \tag{1}$$

*where $I_A$ is the identity operator on $\mathscr{H}_A$.*

Page's theorem therefore says that the *average* distance between a random reduced state $\rho_A(U)$ and the maximally mixed state on $\mathscr{H}_A$, $I_A/d_A$, is bounded by $\sqrt{d_A/d_B}$. The bite of Page's theorem comes when $d_A \ll d_B$, in which case the average distance is very small. Note that this can be the case for qubit systems even when the number of qubits associated with $A$ is only a little smaller than those associated with $B$, as the dimensionality scales exponentially in the number of qubits. In this case, the interpretation of Page's theorem is that the reduced state on a small subfactor of a randomly-chosen pure state is, with high probability, close to the maximally mixed state. Or, in other words, for a Haar-typical state, the reduced state on a small subfactor of Hilbert space is very nearly maximally entangled with its complementary degrees of freedom.

Now consider a CFT in $D$ spacetime dimensions. As mentioned in the introduction, the Haar measure is ill-defined because the Hilbert space of the CFT is infinite-dimensional. Therefore, here we will assume that one also has a regularization of the CFT with the following properties:

- The dimension of its Hilbert space, $\mathscr{H}'$, is finite.

- For a spacelike region $A$ in the CFT, there exists a corresponding subspace $\mathscr{H}'_A \subseteq \mathscr{H}'$, which we can think of as a spatially-local factor of Hilbert space.

- $\log \dim \mathscr{H}'_A$ scales extensively with the volume of $A$. In other words, if the region $A$ in the CFT has a characteristic linear dimension $l$ in some fixed system of coordinates, then

$$\log \dim \mathscr{H}'_A \propto \left(\frac{l}{\epsilon}\right)^{D-1}. \tag{2}$$

Written as such, $\epsilon$ may be interpreted as the (uniform) density of degrees of freedom of $\mathscr{H}'$ in the chosen coordinate system.

For example, $\mathscr{H}'$ could describe a critical spin chain on a periodic lattice of spacing $\epsilon$, where a CFT is recovered in the continuum limit $\epsilon \to 0$ [16–20]. In a similar vein, any tensor network model for holography has a finite-dimensional boundary space by construction [21–25]. Another example is the sort of qubitization of $\mathcal{N} = 4$ super Yang-Mills theory in $D = 4$ dimensions proposed in section 5.3 of ref. [26].

Now that we are armed with a finite-dimensional Hilbert space, let $|\psi'\rangle$ be some Haar-typical state on $\mathscr{H}'$ and consider the reduced state on $\mathscr{H}'_A$. According to Page's theorem, the reduced state on $\mathscr{H}'_A$, call it $\rho'_A$, is very close to being maximally mixed provided that $A$ is a small subregion of the entire CFT. In particular, this means that its von Neumann entropy scales extensively with the volume of $A$, having the same scaling as in eq. (2):

$$S(\rho'_A) \propto \left(\frac{l}{\epsilon}\right)^{D-1}. \tag{3}$$

The way this von Neumann entropy scales is in tension with the Ryu-Takayanagi formula [27], and so a Haar-typical state in $\mathscr{H}'$ like $|\psi'\rangle$ cannot be a good model for a holographic CFT state. Suppose now that the CFT in question has a state $|\phi\rangle$ with a well-defined, asymptotically

AdS geometric dual in $D + 1$ dimensions where we can think of the CFT as living on the bulk geometry's boundary.[4] Per the Ryu-Takayanagi formula, the von Neumann entropy of the reduced state on $A$, call it $\sigma_A$, is given by the area in Planck units of a bulk minimal surface $\tilde{A}$ that is homologous to $A$ and such that $\partial A = \partial \tilde{A}$:

$$S(\sigma_A) = \frac{\text{area}(\tilde{A})}{4G}. \tag{4}$$

The tension comes when $A$ is small enough such that the minimal surface $\tilde{A}$ only probes the near-boundary region, which has asymptotic AdS geometry. Since such $\tilde{A}$ only sees AdS geometry, its area will be subextensive compared to the volume of $A$ in the boundary. For example, when $D = 3$ and when $A$ is a small disk of radius $l$, the area of $\tilde{A}$ scales like $l/\tilde{\epsilon}$, where $\tilde{\epsilon}$ is a UV cutoff in the bulk.[5] In general, the scaling is

$$\text{area}(\tilde{A}) \propto \int_{\tilde{\epsilon}/l}^{1} d\zeta \, \frac{(1 - \zeta^2)^{(D-3)/2}}{\zeta^{D-1}} \leq \int_{\tilde{\epsilon}/l}^{1} \frac{d\zeta}{\zeta^{D-1}}, \tag{5}$$

which is subextensive compared to $(l/\tilde{\epsilon})^{D-1}$. Therefore, Haar-typical states in $\mathscr{H}'$ cannot be good models for typical holographic states, and arguments which depend critically on Haar-typicality will generally not apply to states with classical holographic bulk duals.

## 3 Puzzles resolved and pitfalls espied

Having reviewed Haar-typicality and Page's theorem, we now identify a handful of situations in the literature on holography where intuition from and use of Haar-typical states can be misleading. We also discuss two situations in which the use of Haar-typicality is appropriate.

### 3.1 Measures of holographic states

A current area of research is what fraction of quantum states are allowed to be holographic states. In particular, the measure computed using the entropy cone [30] seems to conflict with a recent numerical study by Rangamani and Rota [31] for reasons that we now clarify.

In their work, Rangamani and Rota study how measures of entanglement and the structure of entanglement among different partitions of a pure state characterize the structure of the state. Given a randomly-chosen pure state of $N$ qubits, they first trace out $k < N$ qubits and then compute various measures of entanglement among partitions of the remaining $N - k$ qubits. For example, one entanglement measure that they check is whether states generated in this way obey monogamy of mutual information (MMI):

$$S_{AB} + S_{BC} + S_{AC} \geq S_A + S_B + S_C + S_{ABC}. \tag{6}$$

In the above, $A$, $B$, and $C$ denotes subsets of the remaining $N - k$ qubits, and all permutations such that $3 \leq |ABC| \leq N - k$ are checked. ($|A|$, $|AB|$, etc. denotes the number of qubits in $A$, $AB$, etc.) MMI is an inequality which holographic states must obey but generic quantum states need not [32]. In their work, it was found that when the method described above is Monte

---

[4]Here we only consider the case where the bulk geometry is stationary, but a generalization to time-dependent geometries using the covariant Hubeny-Rangamani-Takayanagi (HRT) formula [28] is likely possible.

[5]The precise relationship between $\epsilon$ and $\tilde{\epsilon}$ depends on the finite-dimensional theory in $\mathscr{H}'$. For example, ref. [29] argues that a $T\bar{T}$ deformation in a $D = 2$ holographic CFT results in a rigid bulk UV cutoff with specific boundary conditions. Provided that $\epsilon$ and $\tilde{\epsilon}$ do not depend parametrically on the the size of $A$, however, the scalings of $S(\rho_A')$ and $S(\sigma_A)$ may be compared.

Carlo iterated, almost all states generated in this way satisfy MMI. It is further believed that the higher party-number holographic inequalities [30] would be generically obeyed, as well.

The entropy cone measure, by contrast, does not randomly generate states directly, but rather the entanglement entropies $S_A, S_B, S_{AB}, \dots$ directly. Once the requisite entanglement entropies are generated, it is then determined whether they (a) are entanglement entropies that are valid for a quantum state and (b) satisfy the further holographic entanglement entropy inequalities, such as MMI. The ratio of the number of sets of randomly generated entanglement entropies that are consistent with both holography and quantum mechanics as a fraction of the number of such entropies consistent with quantum mechanics is then computed. When using this measure, it is found that just over half of all sets of entropies consistent with quantum constraints obey the holographic inequalities for three parties, and that this fraction appears to fall off rapidly as a function of party number [33].

We stress that Rangamani and Rota's goal was not to compute a measure of holographic states; however, the entropy cone measure is somewhat puzzling in light of how weakly-constraining MMI is in their numerical assays. At this point, it is useful to note that Rangamani and Rota's construction is generic with respect to the Haar measure; the way that the states are constructed there could have been equivalently done via the application of a Haar-random unitary. From the perspective of this measure, if $|A|, |B|, |C| \ll N-k$, then generically we would expect MMI to be not only satisfied, but saturated—not because of any holographic consideration, but because MMI is a balanced inequality and all of the entropies are approximately equal to the logarithms of the dimensions of the states' selfsame Hilbert spaces by Page's theorem.

In general, MMI tends to be satisfied for arbitrary partitions of Haar-typical states, since $S_K = S_{K^c}$, with Page's theorem applied to the complement $K^c$ for any collection of qubits such that $|K| > N/2$. If holographic states cannot be modelled by Haar-typical states in a finite-dimensional Hilbert space, then attempting to conclude which fraction of states can be holographic using a measure that is typical with respect to the Haar measure would yield false positives. The entropy cone measure, by contrast, is unbiased with respect to subregion entropies, as it does not directly generate the states.

## 3.2 Error correction

The assumption that holographic states are well-governed by Page's theorem also appears within the original paper positing the connection between quantum error correction and AdS/CFT [26]. Here, typicality with respect to the Haar measure is used to argue that deletion of arbitrary sets of $l$ qubits in the (qubitized) boundary can be totally corrected given that more than half of the boundary theory is retained. The reason given for this sharp transition is that this is the point at which the deleted portion of the boundary changes between being just less than and just more than half of the number of qubits in the boundary. Then, once the deleted region becomes less than half of the boundary, it becomes exponentially close to maximally mixed. In particular, this is how the relationship between the error correction picture and the entanglement wedge was originally argued. For a more detailed and complete picture of this argument, see [26].

In reality, the constraint provided by Ryu-Takayanagi prevents the deleted region of the boundary theory from being close to maximally mixed, and thus a key portion of the argument above is no longer supported. Indeed, while it is true that a generic state with respect to the Haar measure can be corrected by a typical code with a random $k$ qubit code subspace of $n$ qubits so long as $n - 2l - k \gg 1$, because holographic states are not faithfully modelled by Haar-typical states, this statement has little traction in holography. Thus, this portion of the error correcting picture of holography is not actually suggestive of the entanglement wedge, unless another measure of holographic typicality can be shown to yield similar results. Further works [34, 35] eventually established the relationship between the entanglement wedge and

quantum error correction. Nevertheless, we stress that these Page-type arguments about the relationship between quantum error correction and holography are specious, and must be taken with a large grain of salt, if at all, particularly in the construction of new arguments in this field.

## 3.3 Butterfly effects and shockwaves

Haar-randomness has also been used in the context of ref. [36] to model the behavior of a shockwave acting on the left half of a thermofield double state. To the past of the shockwave, the thermofield double state has a classical bulk black hole geometry. The state is conjectured to have a classical bulk geometry after the shockwave, whose sole effect should be the displacement of the event horizon. This would be difficult to realize by approximating the shockwave as a Haar-random unitary. Based on the previous discussion, acting with a Haar random unitary on (a finite-dimensional regularization of) a CFT state with a well-defined geometric dual would typically create a generic state whose corresponding dual (at least on the left half) would be totally non-classical (in order for boundary subregions to obey Page's theorem). This is undesirable for approximating shockwave geometries, as one loses the ability to reproduce holographic entanglement entropies on spacelike slices of the left region.

## 3.4 Random tensor networks

We note, however, that holographic models can exploit Haar-randomness, as long as the objects that are Haar-random are Haar-random below the curvature scale. For example, the random tensor model of holography [25] has Haar-random unitaries involved in the creation of each individual tensor site. This should not, however, create a global Haar-typical state, as the spatial arrangement of the tensors in the tensor network would provide a structure to the entanglement that is consistent with that of holography. In this way the overall global state avoids being Haar-typical. It is possible that such models may have issues with reproducing physics below the curvature scale, but that is outside of the scope of this work.

## 3.5 Other probability distributions

Similarly, states that are typical with respect to other random state distributions can also serve as good models for holographic states. For example, let $\mathcal{H}_{[E,E+\delta E]}$ denote the subspace spanned by the eigenstates of the Hamiltonian whose energies lie within the range $[E, E+\delta E]$ for some $\delta E \ll E$. A random state drawn with uniform probability from this energy shell (or in other words, sampled from the microcanonical ensemble) typically has a pure-state black hole dual, where the mass of the black hole is set by the energy $E$ [37–39]. In this case, a uniform measure on $\mathcal{H}_{[E,E+\delta E]}$ is admissible because the resulting measure on $\mathcal{H}$ is not itself uniform.

That typical microcanonical states have black hole duals also gives us a nice way to understand, at a heuristic but intuitive level, why random states in the whole CFT should not have good geometric duals (at least for the following notions of randomness and goodness). Consider collating a collection of energy shells with increasing energies. In other words, letting $\mathcal{H}_i \equiv \mathcal{H}_{[E_i,E_i+\delta E]}$, consider the set

$$\mathcal{H}_1 \cup \mathcal{H}_2 \cup \cdots \cup \mathcal{H}_N, \tag{7}$$

with $E_1 < E_2 < \cdots < E_N$. Suppose that we choose a state at random from this set. Heuristically, because the density of states scales exponentially with energy, the random choice of a state will be dominated by the states of the highest energies. In other words, we should expect a randomly chosen state to come from the highest energy shells and, as such, to correspond to a large black hole. However, in the limit where we collate shells that cover the whole Hilbert

space, we should expect a random state to correspondingly describe a black hole that takes up the whole spacetime. There is no asymptotically-AdS geometry, in the sense that boundary-anchored geodesics just skirt along the horizon of the black hole, or equivalently, the boundary of the spacetime, since the black hole is all there is. We are disinclined to think of such a state as being geometric; at the very least, we would not call it a "good" geometric state.

## 4 Conclusion

A randomly-chosen state is probably not holographic. This seems to be generally acknowledged in the holography community, but it can be easy to overlook when tempted with results which stem from Haar-typicality. From this perspective, we clarified some potentially confusing points in the literature. We hope that this will help newcomers to the field to avoid being misled by Haar-random intuition.

## Acknowledgements

We would like to thank Hirosi Ooguri, Daniel Harlow, Massimiliano Rota and Mukund Rangamani for discussions. We would especially like to thank Bogdan Stoica, Sam Blitz, and Veronika Hubeny for collaboration in the early part of this work.

**Funding information** N.B. is supported in part by the DuBridge Postdoctoral Fellowship, by the Institute for Quantum Information and Matter, an NSF Physics Frontiers Center (NFS Grant PHY-1125565) with support of the Gordon and Betty Moore Foundation (GBMF-12500028). A.C.-D. is supported by the Gordon and Betty Moore Foundation through Grant 776 to the Caltech Moore Center for Theoretical Cosmology and Physics. This work is supported by the U.S. Department of Energy, Office of Science, Office of High Energy Physics, under Award Number DE-SC0011632.

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
