# Peer review of "Puzzles and pitfalls involving Haar-typicality in holography"

_SciPost Physics, doi:SciPost Phys. 4, 033 (2018)_

## Round 3 · Referee Report · Anonymous (Referee 1) · 2018-2-8

Strengths

The paper aims to address an interesting question.

Weaknesses

The main statement has to be clarified somewhat.

Report

The authors of this paper address an interesting question: do states which are typical with respect to the Haar measure have a semiclassical geometric holographic dual? They argue that the answer is negative and discuss the relevance of this observation for certain questions in AdS/CFT.

While the discussion in the paper is interesting, I would recommend that the authors address the following issue. I think their statement that "typical Haar states in the CFT do not have a geometric dual" needs to be somewhat modified, or clarified. The Haar measure is defined for finite dimensional Hilbert spaces. The CFT Hilbert space is infinite dimensional, hence to define the Haar measure one first has to regulate it somehow. One natural regularization would be to consider states in the CFT Hilbert space, with energy less than a given value E_0. Then the Haar measure can be defined for all states with energy lower than E_0, or relatedly, the Haar measure can be defined for the microcanonical ensemble of states of energy close to E_0. If the Haar measure is defined in this way, then typical states do have a classical geometric dual, i.e. a big black hole in AdS. This is true no matter how large E_0 is. For this reason, I think it is somewhat misleading to say that Haar-typical states in the CFT do not have a geometric dual.

The authors discuss other possible regularizations, involving a UV cutoff in the theory, and then consider the entire Hilbert space, including states with energies very close to the cutoff. In this case, they correctly point out that typical Haar states would not have geometric dual. However, I think this regularization is less relevant: if we impose such a UV cutoff in the CFT and then study states with energy near that cutoff, then the behavior of the boundary theory in these quantum states would be obviously (and not surprisingly) very different from that of the original CFT for which the AdS/CFT duality is postulated, and it would be more appropriate to think of the breakdown of a geometric dual as an artifact of the regularization, rather than a statement about the original CFT (without a cutoff).

For this reason, I find the claim of the paper that "in AdS/CFT Haar-typical states do not have a geometric dual" potentially misleading. To summarize, defining Haar-typical states in an infinite Hilbert space requires regularization, and there is a very natural regularization (the microcanonical ensemble) according to which Haar-typical state do indeed have a geometric dual. I recommend the authors to make a clarification about this point in their paper.

Requested changes

Described in the report above.

  • validity: ok
  • significance: ok
  • originality: ok
  • clarity: ok
  • formatting: good
  • grammar: excellent

Author:  Aidan Chatwin-Davies  on 2018-02-22  [id 215]

(in reply to Report 1 on 2018-02-08)
Category:
remark
answer to question

We thank the referee for their comments and suggestions. Our response, including changes made which will appear shortly in arXiv v4, is as follows:

Regarding regulators for the CFT, the UV/IR regulator that we use is ubiquitous for information theoretic studies of AdS/CFT. It is intuitive from the perspective of, e.g., CFTs stemming from condensed matter systems, where the CFT is the continuum limit of a many-body system. Moreover, it automatically results in a maximum energy here (for the reason that the Hamiltonian of a finite-dimensional system will have a maximum eigenvalue, up to degeneracy). We added some further discussion about the cutoff in the paragraph above Eq. (2). As the referee has re-iterated, it is indeed crucial to have a finite dimensional Hilbert space for there to be a unique normalized Haar measure.

Note that a maximum energy cutoff alone is not sufficient in general to guarantee a finite-dimensional Hilbert space. For instance, consider a CFT consisting of a massless scalar field in flat spacetime. With a maximum energy (UV) cutoff, there is still a continuum of states between the ground state and any fixed energy (and also within an energy band, for that matter). Similarly, if we instead put the field in a box (i.e., give it an IR cutoff, but no UV cutoff), there is still a countable infinity of field modes. Both a UV and an IR cutoff are necessary to have a finite-dimensional Hilbert space.

We agree with the referee that the Haar measure defined for the microcanonical ensemble can give holographic states with good geometric duals. We believe that this is just a question of nomenclature; we did not refer to it as "the Haar measure" as such because it is not a uniform state measure on the full Hilbert space. We added a brief note in the paragraph at the top of p.3 and we added Sec. 3.5 to clarify this point.

We disagree with the claim that the lack of a geometric dual is an artifact of regularization. In particular, the scaling of subregion entanglement entropy (which, via the Ryu-Takayanagi formula, establishes the necessary condition for there to be a geometric dual that we exploit), does not depend on the regulator (although it is true that the leading term coefficient is not universal). Moreover, even in an unregulated theory, the lack of a geometric dual at any given energy scale is generic. For example, the linear superposition of two black hole microstates with significantly different energies $E_1$ and $E_2$ does not in general correspond to a well defined geometry, and the superposition's energy expectation value can be pushed arbitrarily high by taking black hole states drawn from microcanonical ensembles of arbitrarily high energies. By the same token, these energies can be taken below any cutoff scale. One possible objection is to postulate that all CFT states have quantum gravitational duals in an appropriate quantum gravitational sense. But, in this case, only some of these quantum gravitational duals will have well-defined classical geometric interpretations, and it is these that we focus on.

---

## Round 3 · Referee Report · Anonymous (Referee 2) · 2018-3-19

Strengths

(1) interesting and timely subject
(2) good writing

Weaknesses

(1) the paper only gleans over the most important part of the problem it considers
(2) I think it is incorrect

Report

The manuscript concerns a very interesting and timely question: in holographic CFTs, are Haar-typical states dual to semi-classical geometries? The authors say no, but I think their argument is invalid. I do not recommend publication.

The authors’ main argument can be summarized in one sentence: by Page’s theorem entanglement entropies in a typical state should have volume-law growth yet in states dual to semiclassical geometries the Ryu-Takayanagi proposal mandates that the dominant (UV divergent) contribution to entanglement entropies have area-law growth.

Because Page’s theorem and the definition of the Haar measure both concern finite systems, the crux of the problem lies in explaining how the intuitions of finite systems carry over to CFTs. Unfortunately, the authors don’t spend much time on this question, which I think is the raison d’etre for a paper like this one. The other reviewer already complained that if we consider a band of energies from a CFT then the typical state does represent a semiclassical black hole, which runs contrary to the authors’ claim. So the only way to make the authors’ claim precise is to explain carefully how to appropriately truncate the CFT to a finite system.

The authors’ way to regularize the theory is briefly explained above eq. (2). They truncate the theory by going to a lattice, with an explicit UV cutoff scale epsilon. I am very skeptical about this approach. For definiteness, let’s focus on 1+1 dimensional CFTs. The symmetries of the continuum theory mandate that the entanglement entropies in the vacuum have the Calabrese-Cardy (or Ryu-Takayanagi) form, with a divergent contribution coming from the two endpoints of an interval and a logarithmic dependence on the interval size. When we put the theory on a lattice, these facts continue to be true except for intervals whose size is of the order of the lattice spacing. For such intervals, the distinction between area and volume is meaningless so, in effect, the problem the authors discuss never arises. Even if it were possible to choose “a small subregion A of the entire CFT [so that] the reduced state on A […] is very close to being maximally mixed” AND to distinguish the volume-like dependence of its entanglement entropy from an area law, because a volume-like dependence of entanglement entropy is inconsistent with the symmetries of the theory, I would view this more as an artifact of the regularization than a bona fide feature of an actual conformal field theory.

What is more, there are arguments in the literature which directly contradict the authors’ claim---and which do so on very general grounds. For example, in discussing the Reeh-Schlieder theorem in his recent notes, Witten mentions that the area law dependence of entanglement entropy for small subsystems is universally hard-wired into the structure of the algebra of observables of a QFT (see below eq. 2.21.) This seems to say that there is no limit in which this dependence can be defeated to agree with Page’s theorem, so in fact violations of Page’s theorem are typical (and universal) in QFTs.

I do not recommend the paper for publication. A rewrite that addresses all these concerns would probably turn into a whole other paper.

Requested changes

If the paper is to be published, the authors would have to make the explanation for how to regularize a CFT (to render Page's theorem and Haar measure applicable) the main part of the paper. And I doubt that such a regularization exists. (It would also have to be an "interesting" regularization, not too ad hoc for the problem at hand.)

  • validity: low
  • significance: high
  • originality: high
  • clarity: high
  • formatting: good
  • grammar: perfect

Author:  Aidan Chatwin-Davies  on 2018-04-03  [id 236]

(in reply to Report 2 on 2018-03-19)
Category:
reply to objection

We thank the referee for their report and careful reading of our paper; however, we disagree with their assessment that our analysis is incorrect. We give arguments to rebut their objections below, and we also propose adding a couple of paragraphs to our paper to give better physical intuition for why fully random states should not have good geometric duals.

The referee writes "if we consider a band of energies from a CFT then the typical state does represent a semiclassical black hole, which runs contrary to the authors' claim." This observation does not contradict our claim; as we have indicated in previous correspondence, a uniform measure on an energy band is a very different probability measure on states than the uniform measure on the full Hilbert space. Our claim is only that a typical state with respect to a uniform measure on the whole Hilbert space does not have a good geometric dual. It is perfectly reasonable that there should exist other measures whose typical states do, in fact, have good geometric duals. The uniform measure on an energy shell (i.e., microcanonical ensemble) is one such example.

Next, the referee expresses skepticism about the lattice regularization of a continuum CFT. To this, we point to the extensive literature on the subject. For example, the seminal paper of Vidal, Latorre, Rico, and Kitaev (DOI:10.1103/PhysRevLett.90.227902) establishes a precise correspondence between the scaling of entanglement entropy in critical quantum spin lattices and continuum CFTs (which is the essential result for our paper). The subject of critical lattice systems as CFTs in the continuum limit is further elaborated in hundreds of citing papers and other subsequent literature.

As part of the same train of thought, the referee writes "When we put the theory on a lattice, these facts [the Cardy-Calabrese form of vacuum entanglement entropy] continue to be true except for intervals whose size is of the order of the lattice spacing. For such intervals, the distinction between area and volume is meaningless so, in effect, the problem the authors discuss never arises." First, the problem is not the value of entanglement entropy for boundary subregions whose size is of the order of the lattice spacing. It's a question of how entanglement entropy scales, which is the only invariant question to ask that does not depend on the cutoff (besides the value of universal coefficients) and for which one can envision sending the cutoff to zero. We identify a tension between subextensive (logarithmic in 1+1 dimensions) scaling and volumetric scaling. [As an aside, we also remark that in 1+1 dimensions, and more generally for all odd spatial dimensions, which is when the vacuum entanglement entropy of a region receives a logarithmic correction to area, even the actual values of log(l/a) and (l/a) are quite different when the region size l is of order the cutoff scale a, one being vanishing while the other is not. In this sense, it may be meaningful to distinguish between extensivity and subextensivity in an absolute sense, even at a fixed cutoff scale, but this demands much more than the distinction in scaling that our argument is based on.]

At the end of the same paragraph, the referee writes "because a volume-like dependence of entanglement entropy is inconsistent with the symmetries of the theory, I would view this more as an artifact of the regularization than a bona fide feature of an actual conformal field theory." Area (or log-corrected area) scaling of entanglement entropy is a property of the vacuum and nearby states. Other (excited) states can and do have volumetric entanglement entropy scaling.

Finally, regarding the reference to the notes by Witten (arXiv:1803.04993), the area term is only one term in a series expansion of entanglement entropy, and indeed all states have it (which is why Witten calls it "universal"). Moreover, the area term is not due to the smallness of a region itself, but rather due to the fact that "$\mathcal{H}$ cannot be factored as $\mathcal{H}_{\mathcal{V}} \otimes \mathcal{H}_{\mathcal{V}^\prime}$," as Witten notes. The entanglement entropy picks up an area contribution because a sharp boundary defines the (infinitely) small scale on which any state "looks like" the vacuum. For illustration, refer to Eqs. (1-3) of "Entanglement Entropy and Quantum Field Theory" by Calabrese and Cardy (DOI:10.1088/1742-5468/2004/06/P06002). In all of these equations, which give the entanglement entropy of regions, the area term (which is just a constant in 1+1 dimensions) is there, in addition to the term that scales like the logarithm of the region size. With reference to the previous paragraph, also note that if you expand Eq. (3) in the large temperature limit (corresponding to a large BTZ black hole), the leading contribution to entropy that depends on the region size l scales linearly (i.e., volumetrically) with l.

On a related note, this brings us to a nice way to intuitively understand why fully random states on the full Hilbert space should not have good geometric duals. As has been pointed out, a typical state in an energy shell has a black hole dual geometry, and the energy of the shell corresponds to the size of the black hole. Now imagine taking a shell and widening it to include higher and higher energies, or alternatively collating shells of higher and higher energies. Heuristically, because the density of states scales exponentially with energy, the random choice of a state will be dominated by the states of the highest energies. In other words, we should expect a randomly chosen state to correspond to a large black hole. But, in the limit where the shell grows to encompass the whole Hilbert space, we should expect a random state to correspondingly describe a black hole that takes up the whole spacetime. There is no asymptotically-AdS geometry; boundary-anchored geodesics just skirt along the horizon of the black hole, or equivalently, the boundary of the spacetime, since the black hole is all there is. We are disinclined to think of such a state as being geometric; at the very least we wouldn't call this a "good" geometric dual. Either way, we propose including this short example as an intuitive illustration.

---

## Round 3 · Referee Report · Anonymous (Referee 2) · 2018-4-5

Strengths

(1) interesting and timely subject
(2) good writing

Weaknesses

(1) the paper only gleans over the most important part of the problem.

Report

The issue at hand is what might be meant by Haar typicality in a CFT.

The authors say that thinking about the microcanonical ensemble leads to wrong intuitions because the microcanonical measure (uniform over a band of energies) is different from the Haar measure. But I still don't know what the Haar measure is for a system with an infinite-dimensional Hilbert space.

The authors say that I should think about a lattice regularization of a CFT. I understand and I like the heuristic argument in the last paragraph of their reply: that for fixed UV cutoff in the CFT, almost all states are holographically dual to black holes of size comparable to the gravity IR cutoff, so their geometric interpretation is problematic. But there are two issues: (1) this is a statement about the Haar measure of a lattice regularization of a CFT, not about an actual CFT. (2) this argument is only heuristic.

The biggest worry is (1) above. Consider an infinite sequence of lattice regularizations of a given CFT labeled by the lattice spacing (which asymptotes to zero). Now inspect the corresponding sequence of Haar measures. The limit of such Haar measures is zero on all states, simply because the volume of the Hilbert space blows up. For this reason, I don't know how to make a jump from "the Haar measure of a lattice system" to "the Haar measure of a CFT." The latter concept just doesn't exist.

I suppose the way to make the authors' claim precise is to consider the proportion of the Hilbert space of the lattice system (wrt to the Haar measure), which is occupied by non-geometric states. Call such a quantity $C_a$, where a is the lattice spacing. Then one could argue that $\lim_{a \to 0} C_a = 0$, or at least < 1/2.

To do this, we would have to have a precise definition of $C_a$. This is problem (2) above. It relies, among other things, on understanding how to convert the lattice regularization to an IR gravity cutoff. In holographic 2d CFTs, it was argued that a rigid large scale cutoff in the bulk arises from a T-Tbar deformation of a CFT (1611.03470). So presumably the lattice regularization does not lead to a rigid IR cutoff in the bulk, but something more elusive. For this reason, I'm not sure how to define $C_a$ rigorously.

Requested changes

At the very least, the paper should not refer to "the Haar measure" of a CFT. Just to give an example, the first sentence of the abstract, "Typical holographic states that have a well-defined geometric dual in AdS/CFT are not typical with respect to the Haar measure" presumes the existence of the Haar measure of the full CFT and should therefore be changed.

What would make the paper great is some kind of argument that would lift the statement (which I generally agree with, modulo objection (2) in the report) about the lattice system to the full CFT. At present, I do not know how to take a limit of the lattice claim to get a meaningful claim in the continuum. If the authors manage to provide such an argument, I will be impressed. If not, a carefully presented lattice argument also merits publication.

---

## Round 5 · Author Response

We thank the referees for their reports. Based on their comments and suggestions, we have prepared a revised draft at the request of the editor-in-charge. The revised draft is v5 on the arXiv. Changes, sorted according to referee, are listed below.

---

## Round 5 · List of Changes

Report 1:

  • The changes described in our reply and that we made in v4 were brought forward to v5. The sentence that we added in the introduction was moved to a footnote on p.3, and the new section 5.3 remains in-place.

Report 2:

  • We added the heuristic example to section 5.3. We did not include our detailed rebuttal in the draft, as it is publicly-visible on this submission page.

Report 3:

  • We amended the text throughout to remove any use of the term "Haar-typical state of a CFT". (This includes a revision to the abstract.) We agree with the referee that the term is ill-defined---Haar-typicality indeed requires a finite-dimensional Hilbert space.

  • We further highlight this point in the 6th paragraph of the introduction, and we cleaned up the rest of this paragraph accordingly.

  • Likewise, we re-wrote section 2 starting at the last paragraph on p.4 to be more careful about the finite-dimensional Hilbert space that is to be used in place of the full continuum CFT for defining Haar-typicality. Rather than focus on a specific lattice model, we elected to write down what properties we assume for a finite-dimensional model and we give several examples of models with these properties in the literature, including the critical spin chain.

  • We acknowledged the subtlety that the referee raised regarding the relationship between UV cutoffs in the bulk and boundary in a footnote on p.5.

Further comments:

  • We made small aesthetic formatting changes and corrected minor typos throughout.

  • We also added several references.

---

## Editorial Decision

published